# Prevalence and Characteristics of Female and Male Esports Players among Norwegian Youth: A General Population Study

**DOI:** 10.3390/ijerph21091136

**Published:** 2024-08-28

**Authors:** Stian Overå, Anders Bakken, Christer Hyggen

**Affiliations:** 1Norwegian National Advisory Unit on Concurrent Substance Abuse & Mental Health Disorders, Hospital Innlandet Trust and Norway, 2381 Brumunddal, Norway; stian.overa@sykehuset-innlandet.no; 2Norwegian Social Research (NOVA), OsloMet—Oslo Metropolitan University, 0130 Oslo, Norway; abakk@oslomet.no

**Keywords:** esports, well-being, prevalence, youth, gender

## Abstract

Electronic sports (esports) have evolved into a major cultural phenomenon in the 21st century, mirroring traditional sports with organized, competitive play. This study investigates the prevalence of esports participation and characteristics of esports players among Norwegian adolescents. Using data from the Ungdata survey, a comprehensive web-based survey of Norwegian adolescents that includes questions on demographics, health, physical activity, social relations, and leisure activities such as gaming and esports, we analyzed responses from 70,695 students aged 16–18 years collected during 2021–2023. We conclude that about one out of twenty in the age group of 16–18 years actively engage in esports. The figure conceals significant gender differences: only 0.7% of girls, compared to 8.8% of boys, actively participate in esports. While female esports players differ noticeably from non-players on almost all indicators included in our analyses, the picture among boys reflects more similarities between esports players and other boys, than differences. Among the esports players, females almost consistently reported more negative experiences than males, including less physical socialization, lower self-rated health, higher rates of loneliness and sleeping problems, and a greater difficulty fitting in at school. Despite these negative aspects, esports players maintain close friendships similar to their non-playing peers.

## 1. Introduction

Electronic sports (esports) originated in the arcade era of the 1970s, gained traction during the internet boom of the 1990s, and has since established itself as a significant part of current youth culture in the 2000s [1]. While gaming encompasses a wide range of recreational play across various genres and platforms, esports involves a competitive and organized approach to playing video games [2,3,4,5]. Like traditional sports, esports includes amateur and professional levels, attracting players and fans to popular titles like League of Legends, Fortnite, and Overwatch [3,6]. Esports have extended beyond entertainment, integrating into the social lives and activities of youths [7]. In the context of Norway, Tjønndal and Skauge [1] have termed esports “youth sports 2.0.”, reflecting the tendency for esports in recent years to be integrated into educational programs and local sports clubs [8].

Gaming is one of the most popular activities among youths globally as well as in Norway [9,10]. Esports, an organized and competitive subset of gaming, is one of the fastest growing organized activities for youths in terms of interest and active players. This growth was propelled by the COVID-19 pandemic, when esports gained further popularity in several parts of the world [11,12,13,14]. Norway was no exception. From 2021 to 2024, the “Esports Alliance”, a network of esports clubs in Norway, grew from 18 to nearly 100 member clubs [15]. Moreover, Norway’s official video game strategy [16] supports local initiatives to promote esports, acknowledging the importance of esports as a competitive form of entertainment, cultural platform, and social tool for youth development. 

The increasing popularity of esports has ignited discussions concerning the potential effects on youth, highlighting both positive outcomes—such as fostering social connections [17], reducing stress [18], and developing skills like teamwork, strategic thinking, and hand–eye coordination [13,19,20]—and negative aspects, including physical inactivity [21,22], social isolation, gaming addiction [23], and exposure to harassment [24]. In contrast to traditional sports, where competitions are often separated by gender based on physical disparities, esports frequently include mixed gender tournaments, highlighting the reduced importance of physical qualities in virtual settings [8].

Despite this potential for inclusivity in esports, video games have traditionally been perceived as a “boys’ toy” [25,26,27,28] and an arena dominated by masculine norms and expectations [29,30]. In esports, girls/women are underrepresented both at elite and amateur levels [8,31], and research consistently shows that they face more barriers to participating in esports than their male counterparts. For example, studies have shown that female gamers are more likely to experience exclusion and sexual harassment [32].

To date, much esports research has focused on male professional players, leading to a lack of representation of female perspectives and those of hobbyist players. Moreover, the use of small sample sizes in many quantitative studies has restricted the applicability of their findings [33]. To bridge these significant gaps, the current study employs a large dataset of Norwegian youths (n > 70,000) to investigate the prevalence of esports among Norwegian girls and boys aged 16–18 years. Additionally, we investigate what characterizes young esports players in terms of socio-demographics and well-being, including physical and mental health, social relations, and recreational activities. Since engagement in esports is highly gendered, we explore the potential gender-specific aspects of engaging in esports. In this study, we define esports players as those who exercise/compete esports at least three times per week and typically play video games for at least one hour per day.

## 2. Literature and Research Questions 

The realm of esports research spans various disciplines, but has, until recently, been dominated by business and management perspectives [34,35,36]. Many studies have reported a significant surge in interest and participation in esports in recent years, characterized by increased spectatorship [37], revenue [38], and school-based esports programs [39]. Despite this growth, relatively little is known about the prevalence of esports participation in general youth populations. Additionally, the current research on esports and esports players suffers from a lack of consensus in the operationalization of the phenomenon and how to identify its players [2,3]. However, to separate recreational gamers from esports players, a growing consensus in the literature suggests including criteria such as participation in official organized competitions and the dedication of a certain amount of time with the specific objective of both competing and improving their abilities in esports. It should also be acknowledged that esports can be played at a variety of levels and is not limited in terms of skills of the players, the degree of professionalism, or the level of competition [3,33,40]. The existing literature often focuses on certain aspects of distinct groups of players, from professionals to recreational players, without exploring how prevalent esports are among the youth population or comparing esports players to non-players.

Recent studies from Finland and France have shed some light on the prevalence of esports in general populations. One repeated cross-sectional study investigated the changes in video game habits, including esports, among Finnish high-school students during the COVID-19 pandemic [12]. Data were collected from self-report surveys administered both during and after the spring lockdown in 2020. In the spring of 2020, 16% of the people surveyed reported playing esports games “a lot”. By the fall of 2020, this number increased to 25%. According to researchers, this rise could be explained by the pandemic restrictions that made digital activities more appealing. The study did not examine potential differences in gender or age among the participants. 

The Baromètre France Esports [41], a study conducted annually in France since 2018, provides more detailed insights into the demographics of the esports community. The study collects data from a representative sample of French internet users aged 15 years and older. In the 2023 edition of the study, it was found that 4.5% of the population had registered for a video game tournament over the past year. This group, referred to as “esportifs amateurs”, is predominately composed of young people (91% within the 15–34 age range) and males (93%). However, the recreational player category (“esportifs loisir”), including those 7.2% of the population who play ranked games, allowing to compete with others, but without registering for organized competition, showed a more equal gender split, with 64% of participants being male. These results indicate that the gender balance in competitive esports contexts is more imbalanced than in recreational ones.

Historically, research has highlighted that esports and the broader gaming culture are predominantly male-dominated, significantly affecting women’s participation [29,42,43,44,45,46]. Based on a narrative literature review, ref. [8] identified three central themes that research on gender and esports frequently covers. These themes include (1) the construction of masculinity, where esports environments perpetuate traditional male dominance; (2) gendered expectations, which dictate how individuals should perform and engage based on their gender; and (3) online harassment—often labeled “toxicity” [47]—where women often face disproportionate challenges and hostility. For instance, studies have shown that female players are treated as outsiders [29,43,48] and that female gamers and esports athletes are often met with stereotypes and perceived as less competent than male gamers [24,49,50,51]. In addition, recent research has found that valence and achievement motivation were more relevant for female than for male players and that female players appear more competitive in terms of eagerness to prove themselves in ranked games than male players [31]. Empirical work has also shown that female players have higher perceived stress levels and lower self-reported performance scores than male players [52].

Online gaming and esports have provided a platform for individuals to connect with like-minded people from around the globe. Involvement in esports is reported to provide socializing opportunities and possibilities for collective identity and a feeling of belonging [17]. Research exploring the link between esports participation and loneliness is scarce, and results from research exploring the link between gaming and loneliness are mixed [53]. In general, results indicate that older adolescents and adults who play video games are more prone to experience loneliness than younger players [54]. Online bullying is well-documented among children and youths across several contexts [55], including different gaming communities [56,57,58,59,60], in esports in general—and particularly online bullying directed at female players.

Given the sedentary nature of esports, a considerable amount of research has focused on the physical health of players. It is well-documented that prolonged sitting and lack of physical movement can lead to various negative health outcomes, such as obesity [22], cardiovascular diseases [13], and musculoskeletal problems [61]. The current research on the health habits of esports players has yielded different results regarding whether they are more susceptible to these negative effects than non-players. In a systematic review of the lifestyle behaviors of youth involved in online gaming and esports, either as players or spectators, Chan et al. [21] found that gaming was associated with several negative outcomes, including decreased physical activity, increased body mass index, and a poorer diet. Conversely, in another recent review focusing exclusively on esports players, Monteiro Pereira et al. [62] found that 65–89% of esports players exercise regularly. This is a higher percentage than the general population, also noted in other reviews [35,63]. 

As discussed by the authors of [62], differences in findings across studies in this area might be due to the diversity of game activities included. While some studies group together recreational gaming and esports [21], others focus exclusively on professional and/or educational esports [62,64]. Expanding on the idea of different levels of physical activity in esports players, Trotter et al. [65] found that those ranked in the top 10% of their game tended to engage in more physical activity compared to lower-ranked players, indicating a connection between higher in-game rankings and increased time dedicated to playing esports.

Numerous studies have explored how esports impact well-being and the connection between mental health and video gaming [23,63,66]. While excessive gaming has been linked to depression [67] and social anxiety [68], there is also evidence suggesting that gaming and esports can enhance well-being. For instance, a study in the UK involving 5000 adolescents aged between 10 and 15 years [69] found that those who spent more than one-third of their leisure time gaming exhibited a higher emotional and social well-being than non-gamers. However, children who spent over half their leisure time playing video games reported more emotional and behavioral challenges. The correlations between gaming duration and psychosocial well-being were significant but modest, explaining around 1.6% of the variation. Additionally, research indicates that students engaged in esports have profiles similar to those of their peers. A recent study by Trotter et al. [39] explored the impact of participating in high school esports programs in Australia over time. Their research showed that participating in esports did not affect aspects of mental well-being compared to control groups of the same age. The sleep behavior of gamers and esports players has also garnered considerable research attention. It is well-documented that sleep plays a crucial role in maintaining physical health and overall well-being [70]. Sleep is also key in cognitive performance for esports players, recognized by some as “cognitive athletes” [71]. Several reviews find that esports players often struggle with delayed sleep onset and poor sleep quality [62,63,72]. In a study conducted in Germany, Rudolf et al. [35] found that one out of every six participants, ranging from recreational gamers to professional esports players, reported experiencing poor sleep quality. Research has indicated that extensive gaming, particularly late-night sessions [73], blue-emitting light [74], and stress [75], are factors that can disrupt sleep patterns, leading to insufficient and poor-quality sleep.

Scrutinizing the state of esports research, it is evident that there is a pressing need for more research on how engaging in esports is related to the well-being, social interactions, physical activity, and leisure activities of youths. While the existing literature acknowledges the effects of competitive gaming across these realms, studies often focus on specific areas like mental health [63] or physical activity [72] in isolation. This underscores the importance of taking a more comprehensive approach that considers how social, physical, and mental factors intersect within competitive gaming. Additionally, many studies build on small samples, hindering the broader applicability of the findings [33,76]. Since much research to date focuses on professional esports players, scholars have also stressed the importance of gaining a better understanding of esports involvement across a spectrum of players—ranging from novices to professionals [36,40,77,78]. Finally, as Tjønndal [36] noted, most research on gender and esports comprises qualitative studies. Therefore, conducting large-scale surveys could provide deeper insights into how gender influences individual participation in esports.

In response to the emergence of esports as a global phenomenon, including relatively large proportions of youth populations, existing knowledge, and identified gaps in our knowledge, this study aims to explore how engaging in esports relates to the social interactions, physical health, and overall life satisfaction among 70,695 Norwegian adolescents aged 16–18 years. To accomplish this, the study will investigate the following research questions:How prevalent is playing esports among Norwegian girls and boys in the age group of 16–18 years?What are the differences between girls and boys who are actively engaged in esports and other youths in terms of socio-demographics and leisure activities, social relations, and well-being and health?

## 3. Methods and Measures

### 3.1. Data Collection and Procedure

The analyses in this article utilized data from Ungdata, a standardized survey offered to all Norwegian municipalities and counties by OsloMet and KORUS (ungdata.no). Ungdata aims to map the well-being and leisure activities of young people, including their participation in esports. It is the most comprehensive source of information on adolescent well-being and leisure activities in Norway. Typically conducted triennially since 2010, the survey targets pupils in nearly all lower and upper secondary schools (Grades 8 to 13, ages 13 to 19 years). 

The municipalities themselves commission the Ungdata survey, which includes approximately 150 standardized questions across all surveys, including questions about esports. Each participating municipality can also select additional questions. Ungdata surveys are primarily conducted from January to May, and students complete the questionnaires at school. Participation is voluntary, with non-participants engaging in other school activities. The survey is administered online during school hours under adult supervision. Among participants, nine out of ten completed the last page of the standardized part of the questionnaire. Ungdata is funded by the Norwegian Directorate of Health. For more information, see www.ungdata.no/english/ (accessed on 26 August 2024).

For this study, we used data from the most recent surveys conducted at the upper secondary school level in 2021–2023. The overall response rate was 68%. Our analysis focused on students in the 11th and 12th grades (ages 16–18 years), including responses from 34,656 males and 35,951 females.

### 3.2. Measures

**Esports**. To measure esports engagement, we asked all respondents how often they typically exercised or competed in esports activities, with six response options ranging from “never” to “five or more times a week”. This is consistent with how other types of sports participation are measured in the surveys. To distinguish esports players from casual players or gamers, we set a criterion that esports players should engage in esports activities at least three times a week. Additionally, we set a criterion that they normally played computer games after school at least one hour daily. We established this additional criterion because, upon inspecting the dataset, we found that a certain number of respondents reported being involved in esports without playing computer games.

**Socio-economic status and socio-demographic indicators**. To measure socio-economic status, we asked about parental education, the number of books at home, and four items from the Family Affluence Scale [79]. Based on the average scores of these variables, we constructed a single socio-economic score, and each respondent was placed on a scale of 0–100 [80]. Perceived family poverty was measured by asking “Has your family’s economic situation been good or bad during the past two years?”, with five response options ranging from “always good” to “always bad”. We contrasted those who perceived the family’s economic situation as “mostly bad” or “always bad” with all other adolescents. Immigrant background was measured by asking where the respondent’s parents were born. Those who answered that both parents were born abroad were classified as having immigrant background.

**Leisure time activities**. We utilized three questions from an instrument measuring the frequency of informal leisure time activities over the past week that are common among respondents of this age: “Together with friends at home”, “Hanging out with friends at nights”, and “Been home the whole evening”. Four response options ranged from “None” to “Five times or more”. **Organized leisure activities** were measured by asking participants how many times they had engaged in various organized activities during the last month, including organized sports, youth clubs, religious organizations, cultural organizations, and other organizations. There were four response alternatives, ranging from “none” to “five times or more”. We used the maximum response each respondent selected. The respondents’ **physical activity** was assessed by asking ‘How often do you engage in physical activity that makes you breathe hard or sweat?’, on a six-point scale ranging from “Never” to “At least five times a week”. **Exercise habits** were measured by an instrument measuring the respondent’s frequency of exercising on four different arenas: sports club, gym, personal training, and other types of organized training. Six response options ranged from “never” to “five or more times a week”. **Social media** use was measured by asking how much time the respondent used social media on an ordinary day, with six response alternatives ranging from “none” to “three hours or more”. **Screen time** was measured by a question where students were asked how much time they normally used outside of school on “activities in front of a screen (TV, computer, tablet, smartphone)”, with six response options varying from “No time” to “More than 6 h”.

**Social relations**. **Close friendships** were measured by the following question: “Do you have at least one friend you can trust completely and talk about everything to?”, with the response options ranging from “Yes, for sure”, “Yes, I think so” and “I don’t think so/I don’t have any friends at the moment”. **Fit in among students in school** was measured by a single item taken from an instrument measuring well-being in school. The students were asked to decide whether a statement about whether they fit in among the students at the schools was true, partly true, partly untrue, or not true for them. **Loneliness** was measured by asking whether the students had been bothered by feelings of loneliness during the last week. Four response options were provided, varying from “Not bothered at all” to “Extremely bothered”. **Online bullying** was measured by the following question: “Are you being bullied, threatened or banned online or on your mobile?”, with six response options ranging from “Never” to “Yes, several times a week”.

**Well-being and health**. Life satisfaction was measured by Cantril’s ladder, where the respondents were asked “Imagine a scale from 0 to 10. The top of the scale (10) represents the best possible life for you, and the bottom (0) represents the worst possible life for you. In general, where are you currently on this scale?”. **Subjective well-being** was measured by five items covering both affective and eudaemonic aspects of the respondent’s quality of life. On a five-point scale varying from “Not at all” to “All the time”, respondents were asked to report during the past week whether they had “Been happy”, “Been enthusiastic”, ”Had lots of energy”, “Felt useful”, or “Felt that you master things”. **Self-rated health** was assessed by asking how satisfied the respondents were with their health on a five-point scale ranging from “very unsatisfied” to “very satisfied”. **Future life expectations** were assessed by one item about whether the respondents expected to live a good and happy life. The response options were “yes”, “no”, and “don’t know”. We contrasted those who responded yes with all other respondents. **Depressive symptoms** were measured with five. Items from Kandel and Davies’s Depressive Mood Inventory [81] were used. This measure was derived from the widely used Hopkins Symptom Checklist [82] and assesses depressive symptoms during the preceding week on a four-point scale from “not at all affected” to “extremely affected”. **Sleeping problems** were assessed by one item measuring “Had sleep problems” in the past week, with the same response options as the above measure.

### 3.3. Analyses

To examine the prevalence of esports participation among girls and boys, we conducted a cross-tabulation between gender and the measurement of esports participation. Furthermore, we investigated whether esports players and non-players differed in terms of socio-demographic factors and other indicators of well-being and everyday life by testing for significant differences in mean scores. All tests were conducted separately by gender. We calculated Cohen’s d to enable a standardized comparison across all indicators. In general, Cohen’s d values of 0.2 represent a small effect size, 0.5 a medium effect size, and 0.8 a large effect size. Additionally, we transformed all independent variables into POMP scores [83]. Following the POMP score approach, the variables were rescaled with minimum and maximum possible scores of 0 and 100, respectively. Scores can be interpreted as the percentage of the maximum possible score achievable on the scale.

## 4. Results

Table 1 provides an overview of how youths responded to the question about how often they exercised or competed in esports. Nine out of ten answered that they never or rarely were involved in such activities, suggesting that 10 percent of the youth population reported at least some involvement in esports. Within this group, there were significant differences in frequency. Specifically, 2.2 percent reported involvement less frequently than once a week, while 2.5 percent were engaged in esports once or twice a week. Additionally, 5.5 percent were involved at least three times a week. Moreover, the results show significant gender differences. Among females, 3.1 percent reported at least some involvement in esports, compared to 17.4 percent among males.

In line with recent literature reviews, we distinguished esports players from casual players or gamers. As explained in the Methods Section, we set as a criterion that esports players should exercise/compete esports at least three times a week and typically play computer games at least one hour daily. Using these criteria, where both had to be met, 4.3% of young people were defined as esports players. This applies to 8.8% of boys (n = 3086) and 0.7% of girls (n = 267).

In Table 2, we compared those who were actively engaged in esports and other youths in terms of socio-demographics and various aspects of leisure activities, social relations, and well-being and health. Due to the skewed gender distribution, all analyses were performed separately across gender.

Male esports players do not differ from their non-playing peers in terms of basic socio-demographic indicators, such as socio-economic status, poor household economy, immigrant background, or school grades. Female esports players, on the other hand, are more likely than non-players to come from a poorer socio-economic background, including more likely to experience poor economy in the household and being from an immigrant background. The observed differences are considered small to moderate, but statistically significant.

Male esports players are somewhat less likely to physically socialize with peers and more likely to spend the entire evenings at home than non-players. This trend is also observed among female players, but the differences between female esports players and non-players are considerably more pronounced. Female esports players spend less time socializing with friends physically, either at home or hanging out at night. They spend more time at home and less time socializing digitally on social media compared to non-players.

Male esports players report a substantially higher screen time than non-esports players. However, they differ less in terms of physical activity and exercise. Female esports players also report significantly higher total screen time than non-players, along with lower levels of physical activity and exercise. It is worth noting that, although female esports players report a considerably higher screen time than non-players, they spend less time using social media compared to female non-players. Loneliness is more prevalent among male esports players than non-esports players. Despite this, they experience almost the same level of close friendships and sense of fitting in among students at school as their non-playing peers. Male esports players also experience slightly more online bullying than non-players, but the differences are relatively small. Similar to their male counterparts, female esports players are more likely to experience loneliness compared to non-players, but to a much greater extent. Additionally, female esports players are significantly less likely to feel that they fit in among their peers at school. They also report experiencing online bullying more frequently than non-players, though at the same rate as male esports players.

Male esports players report a lower life satisfaction, subjective well-being, self-rated health, and future life expectations than male non-esports players. In addition, they report higher levels of depressive symptoms and sleep problems. The strength of these associations is considered small. The patterns are similar for females, but stronger, and may be considered at a moderate effect size level.

## 5. Discussion

The aims of the present study were to examine the prevalence of esports participation among Norwegian adolescents and to investigate gender differences in various facets of leisure activities, social relations, and well-being and health among esports players and non-players. In total, the analysis is based on responses from over 70,000 participants aged 16–18 years during 2021–2023, providing novel insights into the extent and influence of esports on youths today.

At present, there are no conventional methods in the research literature for identifying individuals engaged in esports, except by recognizing activities involving competition within organized structures. We opted to map esports involvement by asking respondents how often they participate in training or competition, thus relying on self-reporting, as is normal when measuring participation in other sports or leisure activities. According to this measure, ten percent of Norwegian 16–18-year-olds report engaging in esports. However, the frequency of involvement varied significantly among individuals, with many reporting rare participation.

As our focus was primarily on active esports participants rather than recreational gamers, we defined esports players using a dual criterion approach based on frequency and intensity. The participants were required to participate in esports training/competition at least three times a week and spend a minimum of one hour playing video games daily. Using these criteria, we found that 4.7% met the criteria of esports players.

Our results show a significant gender disparity among the esports players, with 8.8% of boys and only 0.7% of girls meeting both criteria. This uneven gender distribution—93% males and 7% females among the players—is consistent with earlier research that esports are a predominantly male-dominated domain with significant gender disparities [35,41,51].

Several explanations have been proposed to account for this gender gap. As argued by Taylor [29], it may reflect that video games have historically been targeted towards young males, leading esports to be perceived as a masculine activity at the cultural level (see also [43]). This perception can create a «gatekeeping» culture within the esports communities that stigmatizes certain groups, especially females, as “outsiders” [24,45,58]. For example, previous studies have shown that male characters are portrayed four times as often as female characters on video game covers [42], and that games with female characters on the cover result in lower sales figures [44]. Furthermore, while studies have shown that girls engage in gaming as frequently as boys on mobile devices and tablets, boys are overrepresented on platforms (PC and consoles) [25] and in game genres (shooting, strategy, and sports games) that are popular in esports [46]. Researchers have also argued that esports have a diversity problem because online harassment—often termed “toxicity” [47]—disproportionately impacts certain groups. Several studies have found that female gamers are more likely to encounter verbal aggression based on their gender [84,85] and that female players often try to conceal their gender identity to avoid being “exposed” as women to other players [31,59,60].

Another finding from this study is that male and female esports players exhibit slightly different socio-demographic profiles: While male esports players showed small differences from their non-playing peers in terms of household economy and immigrant background, female players reported more disadvantaged backgrounds. These differences, ranging from small to moderate, are significant and may—as we will discuss later—underpin the more negative scores observed in the female players compared to male players across the three life domains investigated in this study: leisure time activities, social relations, and well-being and health.

Despite societal concerns about the sedentary nature of gaming, esports players reported only slightly lower levels of physical activity than non-players. This pattern aligns with recent studies from Norway [74], Germany [35], and Australia [39], highlighting that esports players emphasize physical activity as important to enhance general health. As described by Tjønndal and Skauge [86], in Norway, esports are structured similarly to traditional sports, where physical activity is often integrated as part of schools and sports clubs’ esports programs. One limitation in our study is the lack of information on whether participants are affiliated with any esports clubs or schools. However, it is evident that many esports players maintain lifestyles that are both digitally and physically engaged, consistent with other studies challenging the image of sedentary esports players [22,35,64].

While esports players and non-players do not differ much in their involvement in organized activities, their overall socializing patterns are more home-centered. Esports players, especially females, spend less time than their peers in traditional social settings, such as spending time with friends at home or hanging out at night. Based on results from earlier research on esports [12], this pattern may be related to the time commitment for esports training and matches, which often take place during evenings, when other social activities occur. Esports players of both genders report a notably higher screen time than their non-playing counterparts. Furthermore, while females report more social media use than males, which aligns with previous research from Norway [25], our results reveal an interesting nuance: while female esports players utilize social media less than non-playing females, the opposite trend is true for males. One possible explanation for this divergence is that male players are more inclined than their female counterparts to engage with the esports community through social media, such as following live streams on Twitch, participating in online forums on Reddit, or organizing training sessions on Discord. Additionally, the lower levels of social media use and physical socializing among female esports players could indicate that they experience more social isolation than male players and other girls.

Another related finding in this study was that, while male esports players exhibit social experiences similar to their peers, the situation for females is somewhat different. Encouragingly, most esports players report having close friendships, which is comparable to their non-playing peers. However, loneliness is more prevalent among esports players, particularly among females, who also report significant challenges fitting in among students at school. For male players, while loneliness is slightly higher than among their non-playing counterparts, their sense of fitting in with students at school remains largely unaffected. Building upon findings from other studies, this gender disparity may stem from the fact that gaming culture often serves as a central pivot in males’ friendships and interaction [30,87,88]. For girls, however, finding like-minded gender peers can be more challenging [59]. Some studies have shown that girls “downplay” their interest in gaming to better fit in with their peers and avoid the potential stigma associated with being a “female gamer” [25,89].

As noted above, esports communities have often been described as having a “toxic culture” [47,90], which is particularly hostile towards women and other minorities. In this study, esports players of both genders reported more online bullying than non-players, which could be attributed to the fact that online interaction is an integral part of esports. Interestingly, the results show that male and female players experienced the same level of online bullying. While “online bullying” in the current study includes negative behaviors across various online platforms, the results do not necessarily reflect the specific types of harassment that different genders encounter within gaming environments. For instance, in a qualitative study of Australian esports students aged 18–25 years, Türkay and colleagues [90] found that girls often receive more harassment based on their identity as females, whereas boys primarily receive harassment when they underperform or make mistakes in the game [26,27,32]. If this is a general characteristic in esports, which also translates into our study, it is reasonable to assume that the identity-based harassment faced by females has a more negative impact on their self-esteem and sense of belonging in gaming environments.

Finally, esports players reported worse outcomes compared to non-players on several measures related to well-being and health. However, these effects were more pronounced among female players, who consistently responded lower life and health satisfaction, subjective well-being, future life expectations, as well as more depressive symptoms and sleeping problems. While there is a lack of comparable quantitative studies on gender and esports [36], previous qualitative research suggests that these gender differences may stem from the unique challenges and stressors that female players face in a predominantly male environment [46,88,89,90,91]. Additionally, given that this study finds that female esports players report a more disadvantaged socio-economic background than their male peers, it is also possible that female players bring some of these stressors into the gaming sphere. Studies consistently indicate that socio-economic factors like household economy can contribute to increased stress and other psychological challenges [92,93]. In this research, we lack information regarding potential gender variations in the reasons behind engaging in esports. Nevertheless, it is possible to hypothesize that female players, encountering greater socio-economic obstacles compared to male players, might be more inclined toward seeking escapism through esports participation. In the context of esports, the combination of external socio-economic stressors and gender-related challenges within the gaming community, such as toxicity, might further exacerbate the emotional and mental strain experienced by female players. This dynamic could potentially affect their overall well-being and gaming experience.

Based on this study’s findings, we propose that future research focus on gaining a better understanding of the challenges and stressors faced specifically by females in esports. Additionally, future studies should consider socio-economic factors, as our results indicate a connection between socio-economic background and the well-being and health of esports players.

Furthermore, we recommend that parents, teachers, and other adults take an interest in the passion that young people have for esports. Engaging with their gaming interests can help identify any challenges the young players may encounter, such as isolation or toxic experiences. Furthermore, by promoting a balanced lifestyle that includes physical activity and social interaction, adults can help reduce some of the negative effects associated with excessive screen time. In this regard, the local community can also play an important role. For instance, schools, libraries, and sports clubs could arrange esports events for young people. When planning such initiatives, it is especially important to facilitate participation for girls and young people who are not active in other leisure activities and struggle socially with their peers.

## 6. Conclusions

Using a large-scale study of Norwegian students, we conclude that about one out of twenty in the age group 16–18 years actively engage in esports. This figure conceals significant gender differences: only 0.7% of girls, compared to 8.8% of boys, actively participate in esports. While female esports players differ noticeably from non-players on almost all indicators included in our analyses, the picture among boys reflects more similarities between esports players and other boys than differences. Among the esports players, females almost consistently reported more negative experiences than males, including less physical socialization, lower self-rated health, higher rates of loneliness and sleep problems, and a greater difficulty fitting in at school. This gendered pattern may be explained by the greater social significance and cultural acceptance of esports involvement and gaming skills among boys than among girls. Additionally, the findings suggest that female players come from less privileged socio-economic backgrounds than their non-playing peers and male players. This socio-economic disparity might lead female players to bring other stressors, challenges, and motivations into the esports environment compared to male players. This study provides novel perspectives on the gender-related characteristics of youths in a general population participating in esports and underscores the need for further research and targeted interventions to address the well-being of esports participants, particularly females.

## Figures and Tables

**Table 1 ijerph-21-01136-t001:** Frequency of esports training and competition among males and females.

How Often Do You Exercise or Do the Following Activities: Esports (Train or Compete)?	Males	Females	Total
Never or rarely	82.6	96.9	89.8
Once or twice a month	3.5	0.9	2.2
1–2 times a week	4.1	0.9	2.5
3–4 times a week	4.0	0.7	2.3
5 or more times a week	5.8	0.7	3.2
Total	100.0	100.0	100.0
N=	34,572	35,573	70,145

**Table 2 ijerph-21-01136-t002:** Socio-demographic characteristics and indicators of leisure activities, social relations, and well-being and health among male and female esports players and non-players.

	Males						Females								
	Non-Players		Esports Players				Non-Players		Esports Players				Total Sample		Gender Diff. among Players
	Mean	S.D	Mean	S.D	Cohen’s D	Sig P	Mean	S.D	Mean	S.D	Cohen’s D	Sig P	Mean	S.D	Sig P.
Socio-demographic indicators															
Socio-economic status	65.1	18.2	64.7	17.7	0.02	0.218	66.5	18.5	61.1	18.3	0.30	<0.001	65.8	18.4	0.002
Poor economy at home (%)	4.3	20.3	5.2	22.2	0.04	0.023	5.0	21.9	12.1	32.7	0.33	<0.001	4.7	21.3	<0.001
Immigrant background (%)	13.7	34.4	13.0	33.6	0.02	0.262	13.6	34.3	17.9	38.4	0.13	0.043	13.7	34.3	0.023
Leisure time activities															
Together with friends at home	44.0	33.2	37.7	33.9	0.20	<0.001	47.9	30.6	32.6	32.0	0.48	<0.001	45.6	32.0	0.017
Hanging out with friends at night	39.0	32.6	32.5	32.8	0.20	<0.001	38.2	31.0	25.3	30.9	0.41	<0.001	38.3	31.8	<0.001
At home the whole evening	66.4	28.5	75.3	27.6	0.32	<0.001	67.8	26.1	82.3	25.5	0.53	<0.001	67.5	27.3	<0.001
Organized leisure activities	46.4	43.5	45.4	43.7	0.02	0.203	42.0	42.9	34.8	40.8	0.17	0.007	44.1	43.2	<0.001
Physical activity	73.4	25.4	69.9	25.2	0.14	<0.001	66.1	25.1	60.7	25.5	0.21	<0.001	69.5	25.5	<0.001
Exercise habits	28.6	19.2	26.7	19.3	0.10	<0.001	27.0	17.2	23.9	18.9	0.17	0.004	27.7	18.2	0.022
Social media use	65.3	29.0	67.8	29.0	0.09	<0.001	80.3	23.8	75.0	29.7	0.19	<0.001	73.1	27.5	<0.001
Screen time	71.4	22.4	83.2	18.5	0.54	<0.001	72.8	20.9	87.0	17.2	0.66	<0.001	72.7	21.6	0.001
Social relations															
Close friendships	75.4	34.7	75.8	35.1	0.01	0.562	78.2	33.3	73.5	36.9	0.14	0.021	76.8	34.1	0.31
Fit in with students at school	77.5	26.8	75.1	27.4	0.08	<0.001	71.3	28.6	56.1	33.5	0.54	<0.001	74.1	28.0	<0.001
Loneliness	25.1	31.2	31.5	34.2	0.19	<0.001	37.6	34.2	50.3	36.6	0.38	<0.001	31.8	33.5	<0.001
Online bullying	23.0	12.7	24.3	19.1	0.11	<0.001	22.5	11.2	24.8	20.6	0.19	<0.001	22.8	12.4	0.679
Well-being and health															
Life satisfaction	74.3	17.5	71.6	18.7	0.15	<0.001	67.2	18.5	61.1	20.2	0.33	<0.001	70.5	18.4	<0.001
Subjective well-being	65.6	18.6	63.3	19.6	0.12	<0.001	55.3	18.8	51.3	20.8	0.21	<0.001	60.2	19.4	<0.001
Self-rated health	73.0	29.0	67.0	30.3	0.20	<0.001	62.9	30.9	48.5	33.3	0.47	<0.001	67.5	30.5	<0.001
Future life expectations	71.2	45.3	65.8	47.4	0.11	<0.001	65.8	47.4	46.2	50.0	0.42	<0.001	68.1	46.6	<0.001
Depressive symptoms	29.9	25.0	34.9	26.5	0.18	<0.001	49.2	26.7	59.6	26.7	0.38	<0.001	40.0	27.7	<0.001
Sleeping problems	33.1	31.7	41.0	34.1	0.24	<0.001	42.5	33.3	59.8	35.6	0.52	<0.001	38.3	33.0	<0.001
N=	31,851		3086				36,115		267				71,319		

## Data Availability

The data that support the findings of this study are available from Norwegian Social Research (NOVA), but restrictions apply to the availability of these data, which were used under license for the current study, and so are not publicly available. Data are, however, available from the authors upon reasonable request and with permission of Norwegian Social Research (NOVA).

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
