# Peer review of "Prevalence and Characteristics of Female and Male Esports Players among Norwegian Youth: A General Population Study"

_ijerph, 2024, doi:10.3390/ijerph21091136_

Round 1

Reviewer 1 Report

Comments and Suggestions for Authors

This research is quite interesting but several things need to be improved:

1.      The title should describe the contents and aims of the research.

2.      The abstract does not clearly state the methodology.

3.      Sections 1-3 are poorly written. The idea is not well organized.

4.      The research gap and study aims are repeatedly mentioned in sections 1, 2, and 3. The research gap and the aims of the study should only be written in one section (usually in the last paragraph of the section).

5.      I don’t see the urgency of sections 2 & 3 written in separate sections. It can be merged with section 1. If there’s a need to separate the literature review, then it must be presented in a more comprehensive form.

6.      Changes between paragraphs should be more flowing and smoother.

7.      The authors should add the effect size classification of Cohen D in section 4.2.

8.      The writing of the results shown in percentage form (in sections 5-7) should be accompanied by the number so that readers can get a clearer picture of how many people there are.

Reviewer 2 Report

Comments and Suggestions for Authors

Thank you for the opportunity to review this manuscript. It is a well-designed study that provides valuable insights into the demographics and gaming behaviors of adolescent esports players in Norway.

First, the introduction effectively sets the stage for the study by highlighting the significance of esports in contemporary youth culture, particularly in Norway. The following sections, Literature and Research Gap and Research Gaps and Aim of the Study, provide an adequate review of the relevant literature and clearly articulate the study's objectives. However, some sections could benefit from more concise language to improve readability and flow.

 1. The historical context of esports is well-detailed, but it could be shortened to keep the focus on the current relevance and the study's objectives. That is, what is the connection between the growth of esports and the need for research on adolescent players?

2. The definitions of esports and the distinction from casual gaming are clear. However, the manuscript could benefit from a brief clarification of how "esports" is operationalized in the context of this study earlier in the introduction.

3. Consider highlighting the specific challenges faced by female players earlier in the introduction to set a clearer context for the study's focus on gender differences.

4. The identification of research gaps is thorough. However, the section could be more concise. Focus on the most critical gaps that the current study aims to address.

5. Consistency in terminology needs to be improved. The authors need to Ensure consistent use of terms like "esports," "video games," and "gaming" to avoid any potential confusion.

The methodology needs to be more detailed. It is suggested to have a sub-section for the first paragraph (data collection and procedure). Moreover, as the online survey was carried out, how do you ensure the data quality with a survey consisting of 150 questions? Are there any screening processes to exclude invalid responses?

Finally, the results section is clear and well-interpreted. However, the discussion section could be strengthened by highlighting the theoretical and practical implications. I can expect that this study will contribute to the field of esports industry and provide insights for parents, educators, and policymakers on supporting young gamers.

Comments on the Quality of English Language

The Quality of English Language is good. Minor editing and proofreading work are necessary. 

Round 2

Reviewer 1 Report

Comments and Suggestions for Authors

Thank you for following my suggestions. However, there are still some minor revisions you need to do:

  1. In section 1, I suggest the sentences “In this article, … for at least one hour per day” (lines 38-45) be moved and put into the last paragraph.
  2. I suggest the title of section 2 be changed to “Literature Review and Research Questions” since the aim of the study is already in section 1.
  3. The paragraph in lines 134-136 is only consist of 1 sentence. A paragraph should consist of more than 1 sentence.
  4. The transition between paragraphs before line 134 to line 134 should be more flowing. You can add 1 sentence to connect the paragraphs so that readers can understand the ideas change more fluidly. You can apply this technique to other paragraph changes.

Reviewer 2 Report

Comments and Suggestions for Authors

I would like to thank the authors to address my comments and revise the manuscript. The current version of manuscript is significantly improved. I think the manuscript is good enough to be published.

Comments on the Quality of English Language

Good.
